# Insight into Genetic Mutations of SZT2: Is It a Syndrome?

**DOI:** 10.3390/biomedicines11092402

**Published:** 2023-08-28

**Authors:** Osama Y. Muthaffar, Mohammed M. S. Jan, Anas S. Alyazidi, Taif K. Alotibi, Eman A. Alsulami

**Affiliations:** 1Department of Pediatrics, Faculty of Medicine, King Abdulaziz University, Jeddah 21589, Saudi Arabia; oymuthaffar@kau.edu.sa (O.Y.M.); mmjan@kau.edu.sa (M.M.S.J.); 2Faculty of Medicine, King Abdulaziz University, Jeddah 21589, Saudi Arabia; tmalotibi0001@stu.kau.edu.sa (T.K.A.); erafeealejefi@stu.kau.edu.sa (E.A.A.)

**Keywords:** seizure threshold 2 gene, *SZT2* gene, developmental and epileptic encephalopathy, mTOR, epilepsy, pediatric

## Abstract

Background: The seizure threshold 2 (*SZT2*) gene encodes a protein of unknown function, which is widely expressed, confers a low seizure threshold, and enhances epileptogenesis. It also comprises the KICSTOR protein complex, which inhibits the mTORC1 pathway. A pathogenic variant in the *SZT2* gene could result in hyperactive mTORC1 signaling, which can lead to several neurological disorders. Aim of the study: To review every reported case and present two novel cases to expand the current knowledge and understanding of the mutation. Methods: Whole exome sequencing (WES) was used to identify the novel cases and present their clinical and radiological findings. A detailed revision of the literature was conducted to illustrate and compare findings. The clinical, genetical, neuroimaging, and electrophysiological data were extracted. Results: The study included 16 female patients and 13 male patients in addition to the 2 novel male cases. Eighteen patients had heterozygous mutations; others were homozygous. The majority presented with facial dysmorphism (*n* = 22). Seizures were noted as the predominant hallmark (*n* = 26). Developmental delay and hypotonia were reported in 27 and 15 patients, respectively. The majority of patients had multifocal epileptiform discharges on the electroencephalogram (EEG) and short and thick corpus callosum on the magnetic resonance imaging (MRI). Conclusion: Several promising features are becoming strongly linked to patients with *SZT2* mutations. High variability among the cases was observed. Developmental delay and facial dysmorphism can be investigated as potential hallmarks; aiding clinicians in diagnosing the condition and optimizing management plans.

## 1. Introduction

The human brain serves as the command center and arguably the most important biological system in humans, offering enormous structural complexity [1,2]. Developmental and epileptic encephalopathy (DEE) as well as other central nervous system (CNS) abnormalities obtain substantial heritability, extreme heterogeneity, and genetic predispositions, which are broadly unknown [3,4,5]. This could be largely attributed to the various biological pathways, which could contribute to the pathogenesis of such diseases (i.e., DEE), including deoxyribonucleic acid (DNA) repair, transcriptional regulation, axon myelination, metabolite and ion transport, and peroxisomal function [4]. Recent advances in diagnostic tools, particularly whole exome sequencing (WES), have largely assisted in identifying genetic pathogenic variants, even in rare epileptic phenotypes [6]. Recently, mutations in the seizure threshold 2 (*SZT2*) (MIM 615,463) gene have been described [7], which confer a low seizure threshold and enhance epileptogenesis [7]. The pathogenic variant is located on chromosome 1p34.2 and contains 71 exons [8,9]. The gene was previously known as TIGR, C1orf84, and KIAA0467, and encodes a large protein of unknown function, which is widely expressed in human tissues, and the parietal and frontal cortices, hippocampus, cerebellum, and dorsal root ganglia [7,9,10,11]. *SZT2* is one of four proteins, including KPTN, ITFG2, and C12orf66 comprise the KICSTOR protein complex, which serves as an inhibitor to the mechanistic target of the rapamycin complex 1 (mTORC1) pathway [12,13,14,15], a pathway that mainly regulates cell growth and metabolism [16]. The *SZT2* function within the KICSTOR complex has been suggested to serve as a link between the other three tissues (Figure 1) [12]. Therefore, a *SZT2* variant could result in hyperactive mTORC1 signaling and lead to several neurological disorders [9,17,18]. The aim of this study was to provide further evidence to the currently available literature on the neuro-related clinical presentation of the *SZT2* mutation and systematically analyze the common features that could potentially be described as a syndrome, to aid clinicians in identifying patients with the mutation by emphasizing specific hallmarks related to the *SZT2* mutation. Moreover, we present the case of two siblings, which are carrying a novel homozygous mutation in the *SZT2* gene, with epilepsy and developmental delay, who were born to healthy consanguineous parents carrying a heterozygous missense mutation in the *SZT2* gene (NM_015284.3) and (NM_015284.3), identified through WES. 

## 2. Case Report

### 2.1. Proband I

The proband I is a Saudi child, who is harboring a homozygous c.6113A>Cp. (Tyr2038Ser) (NM_015284.3) variant in the *SZT2* gene, with an uneventful antenatal history, and was a product of spontaneous vaginal delivery (SVD). He is currently 12 years old and presented to our hospital immediately after birth with episodes of GTC seizure that continued to last for two years, with three episodes per day, when on medication. At the age of 2 years, during a routine clinic visit, he was noted as having autistic features and was later diagnosed with autism spectrum disorder (ASD). The patient’s father is a 43-year-old healthy male and a heterozygous carrier of the *SZT2* gene. His mother, and the father’s wife, is a 43-year-old healthy female. The husband and wife are first-degree relatives and there is a positive family history of neurodevelopmental disorders. The patient is currently on valproic acid 600 mg BID, (42 mg/kg/day) and oxcarbazepine (Trileptal) (60 mg/mL, 7 mL bid). His therapeutic range for valproic acid was 83 mcg/mL and the oxcarbazepine level was low. On examination, he was macrocephalic with a high forehead, short philtrum, and triangular face. Moreover, he had a speech delay with limited vocabulary and no sentences. He was noticed to have learning difficulties and visual impairment. His brain magnetic resonance imaging (MRI) came back normal as did his initial electroencephalogram (EEG) but his follow-up EEG, which was done two months ago, demonstrated a few posterior head region sharp waves. The audiology assessment was normal. The patient was recommended to continue on valproic acid (200 mg/mL), same dose, 600 mg BID oxcarbazepine (Trileptal), 420 mg (7 mL) BID, and buccal midazolam 5 mg PRN.

### 2.2. Proband II

The proband II was the younger, and only, sibling. Currently, aged 2 years old with a homozygous c.6113A>Cp. (Tyr2038Ser) (NM_015284.3) variant in the *SZT2* gene. He is a product of SVD by the same parents. He initially presented with a frequent bilateral tonic-clonic seizure, on levetiracetam 60 mg/kg/day (270 mg BID) (2.7 mL BID), carbamazepine 20 mg/kg/day (90 mg BID) (4.5 mL BID), as well as early-onset epileptic encephalopathy and leukoencephalopathy. On examination, he was macrocephalic and hypotonic in all four limbs with normal deep tendon reflexes. He presented with delayed milestones, which included a delay in walking. He started to walk at 2 years old and was still unsteady. He also demonstrated speech delay (non-verbal), although understood simple commands. Moreover, he had a specific learning disability. On inspection, he was noted to have subtle dysmorphic features, including macrocephaly, high forehead, triangular face, prognathism, short philtrum, and posteriorly rotated ears. His EEG and MRI were normal.

## 3. Materials and Methods

### 3.1. Search Strategy

A thorough review of the available literature was conducted among patients reported with *SZT2* mutations. We searched publications from the first published article in 2009 to May 2023 [3,8,9,10,17,19,20,21,22,23,24,25,26,27,28,29,30]. The terms (*SZT2* mutation) and (mTORC2 pathway) were applied and articles were filtered accordingly. The review was carried out in May 2023 and described the clinical, genetical, neuroimaging, and electrophysiological characteristics of 32 patients, as well as presenting 2 additional novel cases in Saudi Arabia with a pathogenic variant in the *SZT2* gene. The literature was obtained from MEDLINE/PubMed, Google Scholar, EMBASE, Scopus, Web of Science, and EBSCOhost databases as well as NEJM, BMJ Case Reports, and Lancet journals. The obtained variables included their clinical data: (A) seizure semiology, (B) epilepsy age of onset, (C) distinguishable clinical features (i.e., facial dysmorphism, intellectual disability, autism, developmental delay, and hypotonia), and (D) management plan. Their genetic background: (A) parental consanguinity status, (B) origin/ethnicity of parents, (C) genetic mutation, (D) patient allele, (E) paternal allele, and (F) maternal allele. In addition to their neuroimaging and electrophysiological findings (i.e., MRI and EEG findings). The study was strengthened according to the narrative review checklist, developed by Green et al., 2006 [31]. The novel cases presented in this study followed the CARE guidelines for case reports [32]. 

### 3.2. Sample Collections

Following appropriate ethical and logistical measures, we obtained genetic samples from patients at King Abdulaziz University Hospital. WES testing was carried out on the two affected siblings and their respective parents. DNA capture probes were used alongside next-generation sequencing (NGS)-based copy number variation (CNV) analysis with Illumina arrays. The coding regions of the gene, 10 bp of flanking intronic sequences, and known pathogenic/likely pathogenic variants within the gene (coding and non-coding) were targeted for analysis. Data analysis, including alignment to the hg19 human reference genome (Genome Reference Consortium GRCh37), variant calling, and annotation was performed using the Torrent Variant Caller (TVC) software. The children’s variant inheritance mode was compared to the parents’ exome sequencing results.

## 4. Results

### 4.1. Literature Review

From the literature search, the study yielded 32 patients with a confirmed pathogenic mutation in the *SZT2* gene [3,8,9,10,17,19,20,21,22,23,24,25,26,27,28,29,30]. We reviewed the demographic, clinical, genetical, neuroimaging, and electrophysiological characteristics of each patient. Additionally, we presented two novel cases that had never previously been described in the literature.

### 4.2. Baseline, Demographic, and Clinical Characteristics

The mean current age of patients was 83.9655 ± 14.282 (±17.01%) months (range: 10–216 months). The study showed a slight female predominance with 16 female patients compared to 13 male patients. Three patients were unknown [25]. The most frequently reported countries included the United States (U.S.) and China, with each reporting six patients. Countries with reported cases are illustrated in Figure 2.

With respect to the patients’ origin/ethnicity, five patients were reported to be Southern Italian, four were Japanese, two were Caucasian, two were from Saudi Arabia, one was Chinese, one was Malaysian, one was Ukrainian, one was Spanish, and one was an Iraqi–Jewish patient, while the others were not reported. Facial dysmorphism was reported among 22 patients, which included a prominent forehead, down slanting, palpebral fissures, ptosis, arched and laterally extended eyebrows, and other distinguished features. Among the seven patients who were reported to have negative dysmorphism, there were three siblings in the United States (*n* = 3) (case numbers 3, 4, and 5), two siblings in Saudi Arabia (*n* = 2) (case numbers 13 and 14), and a single patient in Japan (*n* = 1) (case numbers 8, 9, and 10). Seizures were noted as the predominant hallmark, with 26 patients reporting some form of seizure, while the remaining three cases reported no seizure (case numbers 3, 4, and 5), and another three did not mention them (case numbers 30, 31, and 32). Intellectual disability (ID) was reported among 10 patients. Autism had a slightly lower prevalence, only manifesting in four patients. Developmental delay of any domain was reported in 27 patients with ranging severity across a spectrum of domains. Hypotonia was present in 15 patients. Detailed data on the number of negative and not-mentioned cases are presented in Table 1. 

As for the genetic background of the patients, 18 patients had a heterozygous mutation in the *SZT2* gene compared to 13 homozygous mutations in the same gene. A paternal and maternal allele of the affected gene is thoroughly presented in Table 2.

### 4.3. Seizure Semiology

A spectrum of seizure semiology was reported across the patients. The mean diagnostic age for epilepsy was 24 months (range: 1 day–120 months). Differences between the current age and the age of epilepsy onset were summarized in Figure 3. Focal to bilateral tonic-clonic seizures were reported among most of the patients. Complex partial seizures, generalized motor tonic-clonic seizures, transient seizures following a hypoxic ischemic event, tonic spasm, up-rolling of the eyeballs, status epilepticus, and behavioral arrest were also reported among the patients.

### 4.4. Neuroimaging and Electrophysiological Characteristics

Twenty-six patients were documented to perform EEG. Normal findings were reported in three patients (case numbers 3, 4, and 5). The majority of patients had multifocal epileptiform discharges. Eight patients had slow background activity. Most of the patients with slow background activity had activity originating from the frontotemporal region. The beta wave was recorded in two patients. Lennox–Gastaut-like pattern was seen in one patient (case number 20) and waves were predominantly recorded in the frontal lobes.

Neuroradiological findings: A normal MRI was reported among seven patients (case numbers 3, 4, 5, 15, 16, 18, and 22). A short and thick corpus callosum was a distinguishable radiological sign, which was reported in eight patients (case numbers 1, 2, 8, 11, 20, 23, 27, and 29). Meanwhile, case numbers 24 and 25 reported a thick or abnormally shaped CC with no shortening. However, a thin CC was reported in four cases (case numbers 7, 9, 10, and 26). In six patients, CSP showed persistence on the MRI. Brain atrophy was reported in three patients, including in the hippocampal and cerebellar areas.

### 4.5. Management and Therapeutic Intervention

Management and therapeutic interventions were presented in 21 patients. Multiple ASMs were reported as being administered, which included phenobarbital, levetiracetam, pyridoxine, topiramate, lamotrigine, phenytoin, valproic acid, clobazam, carbamazepine, zonisamide, topiramate, and other ASMs, demonstrated in Table 3, according to each patient. Three cases had eight different types of ASMs administered (case numbers 9,12, and 25); the highest number of ASMs reported in a single patient. Two cases had ten reported medications and nutritional supplements (case numbers 27 and 28). Reported nutritional supplements included potassium bromide, Biotin, or vitamin B7, pyridoxine, folinic acid, and a ketogenic diet. Phenobarbital was reported to be used in 12 patients. Two cases administered ACTH as part of the management plan (cases 19 and 28).

## 5. Discussion

The 71-exon *SZT2* gene encodes a protein of unknown function, which is highly expressed in the CNS, predominately in the parietal and frontal cortices, hippocampus, cerebellum, and dorsal root ganglia [11,33]. In this study, the main purpose was to obtain a broader and updated clinical, genetic, neuroimaging, and electrophysiological insight into a genetic mutation, which has been arguably thought to lead to CNS disorders, namely, developmental and epileptic encephalopathies [5]. Despite studies emphasizing a correlation between acquiring the mutation and having a reduced seizure threshold and enhancing epileptogenesis in mice [33], the human clinical presentation and other phenotypic characteristics remain to be completely elucidated. Furthermore, recent reports suggested that *SZT2* is a component of the KICSTOR complex [12] and dictates control over a protein complex named GATOR, which controls mTORC1 signaling under glucose and amino acid deprivation [13]. The mTORC1 pathway, however, is a master regulator of protein synthesis [35] and integrates nutrient signals to coordinate cell growth and metabolism [13]. The hyperactivated status of mTORC1 eventually led to several clinical presentations, including neurodevelopmental disorders and autism spectrum disorders [36]. Nonetheless, prior research has been performed to emphasize such characteristics [3]; however, new cases have been reported in the literature, including two novel cases in this study. The youngest patient in this study was 10 months old and the oldest was 216 months. The mean age in this study was determined as 83 months in comparison to 92 months in a previous study [3]. Cases with *SZT2* in females became more prevalent, thereby suggesting an increasing incidence of mutation among females where previously a male predominance (59.1% of cases) was described [3]. Furthermore, the gender prevalence shifted despite the addition of two additional male cases in the present study. The majority of patients (*n* = 22) had dysmorphic features. In prior research, it has been described that a high forehead, down slanting palpebral fissures, and ptosis were among the leading dysmorphic features [3], and this study echoes such findings. Epilepsy was reported in nearly 81.25% of the participants in comparison to 86% in a previous study [3]. The semiology was variable; however, focal to bilateral tonic-clonic seizures were notable among participants in a study by Tanaka et al., 2020 [23]. As for the novel cases, their semiology was different, while both siblings had GTC seizures. Nonetheless, complex partial seizures, transient seizures following a hypoxic-ischemic event, tonic spasm, up-rolling of the eyeballs, status epilepticus, and behavioral arrest were also reported among the patients [20,21,28,30,34]. Among the targeted clinical features assessed in this study was the presence of developmental delay, facial dysmorphism, hypotonia, intellectual disability, and autism, which were reported among 84.3%, 78.1%, 46.8%, 31.2%, and 12.5% of the patients, respectively. The extremely high prevalence of developmental delay amplifies prior studies that suggest an active role for *SZT2* in human brain development [9]. Moreover, brain MRI findings showed short and thick CC in most of the patients, whereas the minority of patients had normal MRI findings, including our proband with thin CC, similar to the case reported by Tanaka et al., 2020 [23]. Nonetheless, it is frequently reported that agenesis or dysgenesis of the CC among numerous neurodevelopmental disorders as well as in cases of *SZT2* mutation [9,17,19,20,26,27]. Despite the fact that the majority had multifocal epileptiform discharges on EEG, there was a lack of peculiar electrographic patterns among *SZT2* patients. Findings that were common among *SZT2* can be broadly observed among other DEEs [3]. Similarly, no specific treatment or management modality was described in the literature to markedly halt the disease progression. This includes multiple ASMs administrated and nutritional supplements used as presented in Table 3. Finally, even with the growing number of cases and efforts that have been made to summarize and illustrate a possible syndromic condition, the clinical and electrophysiological presentation remains widely variable. This is due to the nature of the genetic involvement of *SZT2*, which can result in such variation and requires in-depth molecular analysis. Nonetheless, previous speculation on the condition, including drawing a correlation between the mutation and the occurrence of epilepsy, has been less controversial as the evidence increases. Similarly, some clinical and radiological findings are becoming increasingly associated with the condition and are expressed in a broad number of cases, as highlighted in this study.

## 6. Conclusions

In conclusion, several new promising features are becoming strongly linked to patients with *SZT2* mutations as reported cases increase in the literature. A wide spectrum of developmental delay and facial dysmorphism is strongly noted among patients with a mutation in the *SZT2* gene. Despite this, high variability among the cases was observed. However, clinical characteristics that include hypotonia, intellectual disability, and autism can be further investigated as potential hallmarks of the condition, along with developmental delay and facial dysmorphism to a stronger extent; aiding clinicians in suspecting such diagnoses among patients with similar presentations, which could lead to optimization of the management plan. Other relevant findings on the MRI, including a thick and short CC, can be attributed to the hyperactive status of the mTORC1 pathway. EEG remains unspecific with the condition. Overall, there are indeed emerging syndromic features, nonetheless, additional studies are required with expanded sample sizes to draw such a conclusion.

## Figures and Tables

**Figure 1 biomedicines-11-02402-f001:**
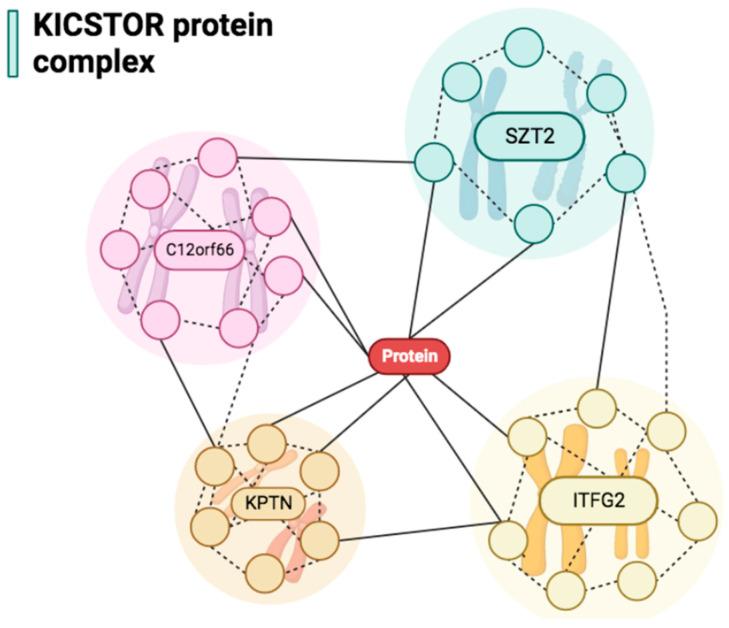
The four proteins (SZT2, KPTN, ITFG2, and C12orf66) comprise the KICSTOR protein complex, which serves as an inhibitor to the mechanistic target of the mTORC1 pathway.

**Figure 2 biomedicines-11-02402-f002:**
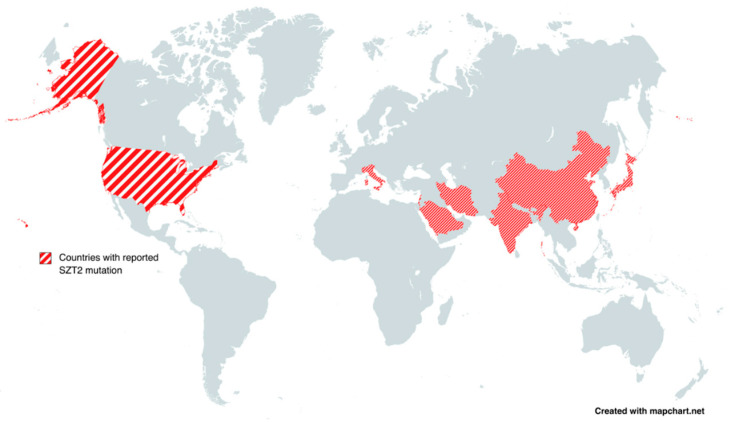
Countries with reported *SZT2* mutations.

**Figure 3 biomedicines-11-02402-f003:**
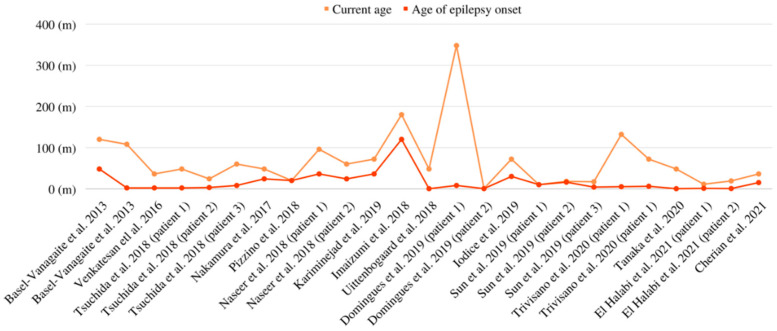
Linear chart for differences between current age and age of epilepsy onset [3,8,9,10,17,19,20,21,22,23,24,25,26,27,28,29,30].

**Table 1 biomedicines-11-02402-t001:** Demographic and baseline clinical characteristics.

Patient No.	Authors	Age	Sex	Reported Country	Facial Dysmorphism	Seizures	Intellectual Disability	Autism	Developmental Delay	Hypotonia
1	Basel-Vanagaite et al., 2013 [9]	10 y	F	Israel	+	+	NM	NM	+ (severe)	+
2	Basel-Vanagaite et al., 2013 [9]	9 y	M	Israel	+	+	NM	NM	+ (severe)	+
3	Falcone et al., 2013 (patient 1) [19]	18 y	M	USA	—	—	+	—	+	—
4	Falcone et al., 2013 (patient 2) [19]	10 y	M	USA	—	—	+	—	+	—
5	Falcone et al., 2013 (patient 3) [19]	7 y	M	USA	—	—	—	—	+ (only speech)	—
6	Venkatesan et al., 2016 [17]	3 y	M	USA	+	+	NM	NM	+	+
7	Vanderver et al., 2016 [33]	7 y	F	USA	+	+	NM	NM	+	NM
8	Tsuchida et al., 2018 (patient 1) [20]	4 y	F	Japan	+	+	+	—	+ (severe)	+
9	Tsuchida et al., 2018 (patient 2) [20]	2 y	M	Japan	+	+	+	—	+ (severe)	+
10	Tsuchida et al., 2018 (patient 3) [20]	5 y	F	Japan	+	+	+	—	+ (severe)	+
11	Nakamura et al., 2018 [26]	4 y	F	Japan	+	+	NM	NM	+	NM
12	Pizzino et al., 2018 [27]	20 m	F	USA	+	+	NM	NM	+	NM
13	Naseer et al., 2018 (patient 1) [8]	8 y	F	Saudi Arabia	—	+	NM	NM	+	+
14	Naseer et al., 2018 (patient 2) [8]	5 y	F	Saudi Arabia	—	+	NM	NM	+	+
15	Kariminejad et al., 2019 [10]	6 y	M	Iran	+ (prominent forehead)	+	NM	+	+	—
16	Imaizumi et al., 2018 [21]	15 y	M	Japan	—	+	NM	+	+	—
17	Uittenbogaard et al., 2018 [28]	4 y	M	USA	—	+	NM	NM	+	+
18	Domingues et al., 2019 (patient 1) [22]	29 y	M	Italy	+	+	+	+	+	—
19	Domingues et al., 2019 (patient 2) [22]	19 y	M	Italy	+	+	+	+	+	—
20	Iodice et al., 2019 [29]	6 y	F	Ukraine	+	+	+	NM	+ (psychomotor)	+ (severe)
21	Sun et al., 2019 (patient 1) [30]	10 m	M	China	+	+	NM	NM	+	+
22	Sun et al., 2019 (patient 2) [30]	18 m	F	China	+	+	NM	NM	+	+
23	Sun et al., 2019 (patient 3) [30]	17 m	M	China	+	+	NM	NM	+	+
24	Trivisano et al., 2020 (patient 1) [3]	11 y	F	Italy	+	+	+	—	—	—
25	Trivisano et al., 2020 (patient 1) [3]	6 y	F	Italy	+	+	+	—	—	—
26	Tanaka et al., 2020 [23]	4 y	F	Japan	+	+	NM	NM	+ (walk and speech)	+
27	El Halabi et al., 2021 (patient 1) [34]	11 m	F	Lebanon	+	+	—	—	+	—
28	El Halabi et al., 2021 (patient 2) [34]	19 m	F	Lebanon	+	+	—	—	+	—
29	Cherian et al., 2021 [24]	3 y	F	India	+	+	—	—	+	+ (truncal)
30	Yang et al., 2022 (patient 1) [25]	NM	NM	China	NM	NM	NM	NM	NM	NM
31	Yang et al., 2022 (patient 2) [25]	NM	NM	China	NM	NM	NM	NM	NM	NM
32	Yang et al., 2022 (patient 3) [25]	NM	NM	China	NM	NM	NM	NM	NM	NM

Abbreviations: F: female, M: male, NM: not mentioned, y: years, m: months. (+) indicates the presence of the variable. (—) indicates the absence of the variable.

**Table 2 biomedicines-11-02402-t002:** Genetic background.

Patient No.	Authors	Consanguinity	Parents Origin	Genetic Mutation	Patient Allele	Paternal Allele	Maternal Allele
1	Basel-Vanagaite et al., 2013 [9]	—	Iraqi–Jewish	Homozygous mutation in the *SZT2* gene	Homozygous for paternally and maternally inherited variant at c.73C<T	c.73C>T p.Arg25*	c.73C>T p.Arg25*
2	Basel-Vanagaite et al., 2013 [9]	—	Spanish	Heterozygous mutation in the *SZT2* gene	Compound of heterozygous for paternally inherited variant at c.1496G>T Maternally inherited variant at c.2092C>T (p.Gln698*)	c.1496G>T p.Gly412Alafs*86	c.2092C>T (p.Gln698*)
3	Falcone et al., 2013 (patient 1) [19]	+ (first degree)	Southern Italy	Homozygous mutation in the *SZT2* gene	Homozygous for paternally and maternally inherited variant at c.4202_4204delTTC	c.4202_4204delTTC (p.Phe1401del)	c.4202_4204delTTC (p.Phe1401del)
4	Falcone et al., 2013 (patient 2) [19]	+ (first degree)	Southern Italy	Homozygous mutation in the *SZT2* gene	Homozygous for paternally and maternally inherited variant at c.4202_4204delTTC	c.4202_4204delTTC (p.Phe1401del)	c.4202_4204delTTC (p.Phe1401del)
5	Falcone et al., 2013 (patient 3) [19]	+ (first degree)	Southern Italy	Homozygous mutation in the *SZT2* gene	Homozygous for paternally and maternally inherited variant at c.4202_4204delTTC	c.4202_4204delTTC (p.Phe1401del)	c.4202_4204delTTC (p.Phe1401del)
6	Venkatesan et al., 2016 [17]	NM	NM	Heterozygous mutation in the *SZT2* gene	Compound of Heterozygous for Paternally inherited variant at c.3509_3512delCAGA (p.T1170RfsX22) Maternally inherited variant at c.9703 C>T (p.R3235X)	c.3509_3512delCAGA (p.T1170RfsX22)	c.9703C>T (p.R3235X)
7	Vanderver et al., 2016 [33]	NM	NM	*SZT2* gene	NM	c.5499del (p. Phe1834Serfs*47)	c.6916G>A (p. Gly2306Arg)
8	Tsuchida et al., 2018 (patient 1) [20]	—	Japanese	Heterozygous mutation in the *SZT2* gene	Compound of heterozygous for paternally inherited variant at c.3700_3716del:p.(Asn1234Alafs*35) Maternally inherited variant at c.5482del:p.(Gly1829Valfs*52)	c.3700_3716del (p.Asn1234Alafs*35)	c.5482del (p.Gly1829Valfs*52)
9	Tsuchida et al., 2018 (patient 2) [20]	—	Japanese	Heterozygous mutation in the *SZT2* gene	Compound of heterozygous for paternally inherited variant at c.3947dup:p.(Glu1317Glyfs*4) Maternally inherited variant at c.2929+1G>A: p.(Leu939Aspfs*19)	c.3947dup (p.Glu1317Glyfs*4)	c.2929+1G>A (p.Leu939Aspfs*19)
10	Tsuchida et al., 2018 (patient 3) [20]	+	Malaysian	Heterozygous mutation in the *SZT2* gene	Compound of heterozygous for paternally inherited variant at c.7303C>T:p.(Arg2435Trp) Maternally inherited variant at c.8162C>G:p.(Ser2721Cys)	c.7303C>T (p.Arg2435Trp)	c.8162C>G (p.Ser2721Cys)
11	Nakamura et al., 2018 [26]	—	Japanese	Heterozygous mutation in the *SZT2* gene	Compound heterozygous for two paternally inherited variants at c.4181C>T (p.Pro1394Leu) and c.2930-17_2930-3delinsCTCGTG Maternally inherited variant at c.8596dup (p.Tyr2866Leufs*42)	c.4181C>T (p.Pro1394Leu) and c.2930-17_2930-3delinsCTCGTG	c.8596dup (p.Tyr2866Leufs*42)
12	Pizzino et al., 2018 [27]	NM	NM	Heterozygous mutation in the *SZT2* gene	Compound heterozygous for paternally inherited variant at c.5499delC (p.Phe1834Serfs*47) Maternally inherited variant at c.6916G>A (p.Gly2306Arg)	—	c.6916G>A (p.Gly2306Arg)
13	Naseer et al., 2018 (patient 1) [8]	+ (first degree)	Saudi Arabian	Homozygous mutation in the *SZT2* gene	Homozygous for paternally and maternally inherited variant at c.9368G>A	c.9368G>A (p.Gly3123Glu)	c.9368G>A (p.Gly3123Glu)
14	Naseer et al., 2018 (patient 2) [8]	+ (first degree)	Saudi Arabian	Homozygous mutation in the *SZT2* gene	Homozygous for paternally and maternally inherited variant at c.9368G>A	c.9368G>A (p.Gly3123Glu)	c.9368G>A (p.Gly3123Glu)
15	Kariminejad et al., 2019 [10]	+	NM	Homozygous mutation in the *SZT2* gene	Homozygous for paternally and maternally inherited variant at NM_015284.3: c.7442G>A (p.Cys2481Tyr)	c.7442G>A (p.Cys2481Tyr)	c.7442G>A (p.Cys2481Tyr)
16	Imaizumi et al., 2018 [21]	—	Japanese	Homozygous mutation in the *SZT2* gene	Homozygous for paternally and maternally inherited variant at c.6553C>T (p.Arg2185Trp)	c.6553C>T (p.Arg2185Trp)	c.6553C>T (p.Arg2185Trp)
17	Uittenbogaard et al., 2018 [28]	NM	NM	Heterozygous mutation in the *SZT2* gene	Compound of heterozygous for paternally inherited variant at c.5949_5951del TGT (p.Val1984del) Maternally inherited variant at c.5174C>T (p.A1725V)	c.5949_5951del TGT (p.Val1984del)	c.5174 C>T (p.A1725V)
18	Domingues et al., 2019 (patient 1) [22]	—	Caucasian	Heterozygous mutation in the *SZT2* gene	Compound of heterozygous for paternally inherited variant at c.6553C>T (p.Arg2185Trp) Maternally inherited variant at c.498G>T (p.Gln166His)	c.6553C>T (p.Arg2185Trp)	c.498G>T (p.Gln166His)
19	Domingues et al., 2019 (patient 2) [22]	—	Caucasian	Heterozygous mutation in the *SZT2* gene	Compound of heterozygous for paternally inherited variant at c.6553C>T (p.Arg2185Trp) Maternally inherited variant at c.498G>T (p.Gln166His)	c.6553C>T (p.Arg2185Trp)	c.498G>T (p.Gln166His)
20	Iodice et al., 2019 [29]	NM	Ukrainian	Heterozygous mutation in the *SZT2* gene	Compound heterozygous for paternally inherited variant at c.3632G>A (p.Arg1211Gln) Maternally inherited variant at c8435delC (p.Ser2812Leufs*41)	c.3632GNA (p.Arg1211Gln)	c8435delC (p. Ser2812Leufs*41)
21	Sun et al., 2019 (patient 1) [30]	—	NM	Heterozygous mutation in the *SZT2* gene	Compound heterozygous for paternally inherited variant at c.1626 + 1G>A: splicing Maternally inherited variant at c.5772dupA (p.C1924 fs)	c.1626 + 1G>A: splicing	c.5772dupA (p.C1924 fs)
22	Sun et al., 2019 (patient 2) [30]	—	NM	Heterozygous mutation in the *SZT2* gene	Compound heterozygous for paternally inherited variant at c.1626 + 1G>A: splicing Maternally inherited variant at c.5772dupA (p.C1924 fs)	c.1626 + 1G>A: splicing	c.5772dupA (p.C1924 fs)
23	Sun et al., 2019 (patient 3) [30]	—	Chinese	Heterozygous mutation in the *SZT2* gene	Compound heterozygous for paternally inherited variant at c.4209C>A (p. C1403X,1973) Maternally inherited variant at c.7307_7308insG (p.A2436fs*22)	c.4209C>A (p.C1403X,1973)	c.7307_7308insG (p.A2436fs*22)
24	Trivisano et al., 2020 (patient 1) [3]	+	Southern Italy	Homozygous mutation in the *SZT2* gene	Homozygous for paternally and maternally inherited variant at c.7825TNG;p.(Trp2609Gly)	c.7825T>G p.(Trp2609Gly)	c.7825T>G p.(Trp2609Gly)
25	Trivisano et al., 2020 (patient 1) [3]	+	Southern Italy	Homozygous mutation in the *SZT2* gene	Homozygous for paternally and maternally inherited variant at c.7825TNG;p.(Trp2609Gly)	c.7825T>G p.(Trp2609Gly)	c.7825T>G p.(Trp2609Gly)
26	Tanaka et al., 2020 [23]	—	NM	Heterozygous mutation in the *SZT2* gene	Compound heterozygous for paternally inherited variant at c.2798C>T (p.Ser933Phe) Maternally inherited variant at c.4549C>T (p.Arg1517Trp)	c.2798C>T (p.Ser933Phe)	c.4549C>T (p.Arg1517Trp)
27	El Halabi et al., 2021 (patient 1) [34]	+ (first degree)	NM	Homozygous mutation in the *SZT2* gene	Homozygous for Paternally and maternally inherited variant at c.82C>T (p.Arg28*)	c.82C>T (p.Arg28*)	c.82C>T (p.Arg28*)
28	El Halabi et al., 2021 (patient 2) [34]	+ (first degree)	NM	Homozygous mutation in the *SZT2* gene	Homozygous for paternally and maternally inherited variant at c.82C > T p. (Arg28*)	c.82C>T (p.Arg28*)	c.82C>T (p.Arg28*)
29	Cherian et al., 2021 [24]	—	NM	Homozygous mutation in the *SZT2* gene	Homozygous for paternally and maternally inherited variant at (p.Asp2440ArgfsTer18)	p.Asp2440ArgfsTer18	p. Asp2440ArgfsTer18
30	Yang et al., 2022 (patient 1) [25]	NM	NM	Heterozygous mutation in the *SZT2* gene	Compound heterozygous for inherited variants at c.2887A>G/c.7970G>A	NM	NM
31	Yang et al., 2022 (patient 2) [25]	NM	NM	Heterozygous mutation in the *SZT2* gene	Compound heterozygous for inherited variants at c.3508A>G/c.7936C>T	NM	NM
32	Yang et al., 2022 (patient 3) [25]	NM	NM	Heterozygous mutation in the *SZT2* gene	Compound heterozygous for inherited variants at c.2489G>T/c.8640_8641insC	NM	NM

Abbreviations: NM: not mentioned. (+) indicates the presence of the variable. (—) indicates the absence of the variable.

**Table 3 biomedicines-11-02402-t003:** Seizure semiology, neuroradiological, and electrophysiological characteristics.

Patient No.	Authors	Epilepsy Onset	Seizure Semiology	EEG Findings	MRI Findings	Management
1	Basel-Vanagaite et al., 2013 [9]	4 y	Focal to bilateral tonic-clonic seizures	Generalized epileptic discharges	Short and thick CC and persistent CSP	NM
2	Basel-Vanagaite et al., 2013 [9]	2 m	Generalized motor tonic-clonic seizures	Slow background and multifocal spikes in either hemisphere	Short and thick CC and persistent CSP	NM
3	Falcone et al., 2013 (patient 1) [19]	NM	NM	Normal	Normal	NM
4	Falcone et al., 2013 (patient 2) [19]	NM	NM	Normal	Normal	NM
5	Falcone et al., 2013 (patient 3) [19]	NM	NM	NM	Normal	NM
6	Venkatesan et al., 2016 [17]	2 m	Generalized motor tonic-clonic seizures	Multifocal epileptiform discharges consistent with a diffuse epileptogenic encephalopathy	Right periventricular heterotopia	Phenobarbital (5.8 mg/kg/day), Levetiracetam (56.1 mg/kg/day), Pyridoxine (17 mg/kg/day), Topiramate (2.8 mg/kg/day), Lamotrigine (4.9 mg/kg/day), and Divalproex (17.4 mg/kg/day)
7	Vanderver et al., 2016 [33]	NM	NM	NM	Deficits in myelination, mild cerebellar atrophy, and volume loss of the CC and FL	NM
8	Tsuchida et al., 2018 (patient 1) [20]	2 m	Focal to bilateral tonic-clonic seizures	Slow, 4–5 Hz background activity with spike waves in the left frontocentral areas	Short and thick CC	Phenobarbital, Phenytoin, and potassium bromide
9	Tsuchida et al., 2018 (patient 2) [20]	3 m	Seizures presented as either with cyanosis and stopping motion, or as adversive seizures with fencing posture	Rhythmic fast waves, low amplitude fast waves, rhythmic slow waves, bursts of spikes, and sharp waves	Diffuse brain atrophy, thinned corpus callosum, and a persistent CSP	Valproic acid, Clobazam, Phenobarbital, Carbamazepine, Phenytoin, Zonisamide, Topiramate, and Levetiracetam
10	Tsuchida et al., 2018 (patient 3) [20]	8 m	Tonic spasm and up-rolling of the eyeballs	Intermittent epileptic discharges over both posterior quadrants. These involved all regions, except the central and anterior areas with poorly organized background	Dilated lateral and third ventricles, and a thin corpus callosum	Clobazam
11	Nakamura et al., 2018 [26]	2 y	Focal to bilateral tonic-clonic seizures	Focal epileptic discharges and background slowing	Short and thick CC	Valproate
12	Pizzino et al., 2018 [27]	20 m	Focal to bilateral tonic-clonic seizures	Multifocal epileptiform discharges	Increasing size and loss of normal posterior pituitary signal	Lamotrigine, Levetiracetam, Phenytoin, Phenobarbital, Topiramate, Rufinamide, Midazolam, and ketogenic diet
13	Naseer et al., 2018 (patient 1) [8]	3 y	NM	Background of predominately beta wave rhythm. Paroxysmal epileptic discharge, mainly on the right central hemisphere	Prominent extra-axial cerebrospinal fluid space with wide Sylvian fissure	NM
14	Naseer et al., 2018 (patient 2) [8]	2 y	NM	Background of predominately beta wave rhythm. Paroxysmal epileptic discharge, mainly on the right central hemisphere	Prominent extra-axial cerebrospinal fluid space with wide Sylvian fissure	NM
15	Kariminejad et al., 2019 [10]	3 y	Tonic-clonic seizures	Normal	Normal	Diazepam (IM injection), Phenobarbital (90 mg/bd), Sodium valproate (400 mg/bd), and Risperidone (5 mg/daily)
16	Imaizumi et al., 2018 [21]	10 y	Complex partial seizures	Multifocal spikes or spikes and waves, predominantly in the frontal lobes	Normal	Unspecified antiepileptic medications
17	Uittenbogaard et al., 2018 [28]	1 d	Transient seizures following a hypoxic ischemic event	Excessive left and right frontal sharp wave discharges were indicated	The involvement of the globus pallidum	Phenobarbital and Levetiracetam
18	Domingues et al., 2019 (patient 1) [22]	8 m	Focal motor to bilateral tonic-clonic seizures	NM	Normal	Sodium valproate, Vigabatrin, Valproate, and Lacosamide
19	Domingues et al., 2019 (patient 2) [22]	4 d	Focal and focal-to-generalized tonic seizures	Sub-continuous bifrontal sharp waves and multifocal epileptiform discharges	Right frontal polymicrogyria	Phenytoin, Carbamazepine, Clobazam, Topiramate, Phenytoin, Zonisamide, Rufinamide, Lacosamide, and ACTH
20	Iodice et al., 2019 [29]	30 m	Cluster of afebrile focal-clonic seizures	Epileptic encephalopathy with a Lennox–Gastaut-like pattern	Thick and short CC and right hippocampal atrophy	Levetiracetam, Valproic acid, Rufinamide, and Clobazam
21	Sun et al., 2019 (patient 1) [30]	10 m	Focal tonic or clonic seizures; status epilepticus	Slow background, epileptic discharges originating from L medial temporal area on ictal EEG	Enlarged ventricle and delayed myelination in the terminal zone	Levetiracetam, Oxcarbazepine, and Phenobarbital
22	Sun et al., 2019 (patient 2) [30]	16 m	Focal or generalized tonic seizures; status epilepticus	Slow background, multifocal discharges on interictal EEG, epileptic discharges originating from L temporal area on ictal EEG	Normal	Valproic acid, Nitrazepam, and Topiramate
23	Sun et al., 2019 (patient 3) [30]	4 m	Focal or generalized tonic seizure; status epilepticus	Slow background, multifocal discharges with partial generalization	Subependymal nodules, shortened corpus callosum, enlarged ventricles, and widened cavum septum pellucidum	Levetiracetam, Topiramate, Clonazepam, and Valproic acid
24	Trivisano et al., 2020 (patient 1) [3]	5 m	Focal to bilateral tonic-clonic seizures, at times with secondary generalization	Bilateral temporal theta waves and rare spikes in right temporal	Thick CC abnormally shaped	Phenobarbital and Carbamazepine
25	Trivisano et al., 2020 (patient 1) [3]	6 m	Focal to bilateral tonic-clonic seizures, at times with secondary generalization during sleep	Altered background, left temporal, right C–P spikes, and slow waves	CC abnormally shaped	Carbamazepine, Levetiracetam, Valproate, Phenobarbital, Clobazam, Oxcarbazepine, Rufinamide, and Lacosamide
26	Tanaka et al., 2020 [23]	4 d	Focal seizures followed by tonic or clonic seizures affecting one or both limbs on the left side; occasionally, bilateral tonic-clonic seizures	Multifocal epileptiform discharges	Thin CC, persistent CSP, and atrophy	Phenobarbital, Midazolam, and Carbamazepine (5 mg/kg/day)
27	El Halabi et al., 2021 (patient 1) [34]	1 m	Staring, blinking, and flushing	Multifocal independent spikes	Short CC of normal thickness and persistent CSP	Phenytoin, Carbamazepine, Oxcarbazepine, Levetiracetam, Clonazepam, Valproate, potassium, bromide, Biotin or vitamin B7, Pyridoxine, and Folinic acid
28	El Halabi et al., 2021 (patient 2) [34]	2 w	Behavioral arrest, increased tone, grimacing, and flushing	Generalized slowing, multifocal independent spikes with ictal activity originating independently from the right frontotemporal, left or bilateral temporal, and on occasions, the ictal discharge propagated from one hemisphere to the other, hypsarrhythmia	Persistent CSP with a CC of normal length and thickness	Levetiracetam, Valproate, Phenobarbital, Clonazepam, Lacosamide, ACTH, potassium bromide, Biotin or vitamin B7, Pyridoxine, and Folinic acid
29	Cherian et al., 2021 [24]	15 m	Focal seizures with impaired awareness (hypomotor events) followed by generalized tonic seizures	Multifocal epileptiform abnormalities, which were maximum bifrontal and paucity of generalized discharges; generalized paroxysmal fast activity (GPFA) or burst attenuation (BA) pattern were conspicuously absent	Thickened, dysmorphic CC with abnormal CSP	Sodium, valproate, Phenobarbitone, Levetiracetam, Clobazam, Perampanel, and Zonisamide
30	Yang et al., 2022 (patient 1) [25]	NA	NA	NA	NA	NA
31	Yang et al., 2022 (patient 2) [25]	NA	NA	NA	NA	NA
32	Yang et al., 2022 (patient 3) [25]	NA	NA	NA	NA	NA

Abbreviations: NM: not mentioned; NA: not applicable; y: years, m: months, w: weeks, d: days. (+) indicates the presence of the variable. (—) indicates the absence of the variable.

## Data Availability

The datasets generated or analyzed during the current study are available from the corresponding author upon reasonable request.

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
