# Peer review of "Insight into Genetic Mutations of SZT2: Is It a Syndrome?"

_biomedicines, 2023, doi:10.3390/biomedicines11092402_

Round 1
Reviewer 1 Report
In Muthaffar et al., the authors describe the genetics and clinical presentations of 32 patients with mutations in the SZT2 gene, as well as two novel cases identified by the authors. Several traits stand out as hallmarks of a possible syndrome, including seizures, facial dysmorphism, and intellectual or developmental delays, but there remains substantial variation in how these patients present. For example, several patients were found to have a short, thick corpus callosum, while others were found to have a thin corpus callosum. This is indicative of the lack of clarity on this subject. The authors state that they cannot conclude whether SZT2 mutations cause a specific syndrome, and this paper does not go further than serving as a summary of existing cases and their characteristics.
In general terms, the paper would benefit from deliberate descriptions of the specific clinical definitions that are used. This will help avoid confusion that could result from variation in clinical definitions. T
he manuscript would benefit from thorough copy editing. Here is some examples:
Line 13: “inhibit” à “inhibits”
L. 21: should be “mutations,” (plural with a comma)
L. 43: should be “enhances” or “enhanced”
L. 54: should be “currently”
L. 56: should be “a syndrome”
L. 59: should it be a heterozygous missense mutation?
L. 74: sentence doesn’t really make sense
L. 92: “He is known case of neonatal teeth…” sentence needs to be rewritten
L. 115: respectively?
Table 2: text is spilling into 2 lines
L. 180: should be “most of the patients”
Author Response
Thank you dear reviewer for your valuable insight. Your comments are much appreciated and contributed to improving our work. Kindly note that all comments were addressed. Also, we revised the whole manuscript using language improvement tools that detected very minor linguistic errors which were all corrected. We believe the paper is better in its current form and we hope we qualify for an " English language fine. No issues detected " choice by you.
Line 13: “inhibit” à “inhibits”
- Modified into plural form.
- 21: should be “mutations,” (plural with a comma)
- Modified into plural form and comma added.
- 43: should be “enhances” or “enhanced”
- Corrected into "enhances".
- 54: should be “currently”
- Change to currently.
- 56: should be “a syndrome”
- Corrected
- 59: should it be a heterozygous missense mutation?
- Thank you for your valuable comment and we apologize for this error. Corrected after double-checking the WES testing results.
- 74: sentence doesn’t really make sense
- We rephrased it to enhance clarity and meaning.
- 92: “He is known case of neonatal teeth…” sentence needs to be rewritten
- We re-wrote this segment and merged it to the examination part for a smoother construction. I hope it's clearer now and let me know of any suggestions.
- 115: respectively?
- Removed.
Table 2: text is spilling into 2 lines
- This was made by the journal and we hope during publication this issue will be resolved while drafting the final form of the research.
- 180: should be “most of the patients”
I believe you're referring to this line " Focal to bilateral tonic-clonic seizures were reported in 13 patients". However, 13 could be controversial in most patients as the total number is 32 despite the fact that this is the most repeated form of seizure. But will modify the sentence according to your suggestion.
Kind regards,

Reviewer 2 Report
An interesting and well written paper. I suggest the authors to be more explicit in the conclusion by emphasizing the main clinical features of the proposed STZ2 syndrome, indicating the need for specific genetic study in patients with with clinical criteria for the diagnosis of suspicion for this syndrome. In addition, some of the abbreviations should be defined in the text
Author Response
Thank you dear reviewer for your valuable insight. Your comments are much appreciated and contributed to improving our work. Kindly note that all abbreviations were defined, thus, addressing your comment. Also, we revised the whole manuscript using language improvement tools that detected very minor linguistic errors which were all corrected. We believe the paper is better in its current form and we hope we qualify for a " English language fine. No issues detected " choice by you.
Kind regards,
